# Enniatin A1, A Natural Compound with Bactericidal Activity against *Mycobacterium tuberculosis* In Vitro

**DOI:** 10.3390/molecules25010038

**Published:** 2019-12-20

**Authors:** Gaoyan Wang, Wenqi Dong, Hao Lu, Wenjia Lu, Jiajia Feng, Xiangru Wang, Huanchun Chen, Manli Liu, Chen Tan

**Affiliations:** 1State Key Laboratory of Agricultural Microbiology, College of Veterinary Medicine, Huazhong Agricultural University, Wuhan 430070, Hubei, China; 97wgy@webmail.hzau.edu.cn (G.W.); 13137204926@163.com (W.D.); sdluhao521@163.com (H.L.); hzaulwj1995@163.com (W.L.); 19fj@webmail.hzau.edu.cn (J.F.); wangxr228@mail.hzau.edu.cn (X.W.); chenhch@mail.hzau.edu.cn (H.C.); 2Hubei Biopesticide Engineering Research Centre, Hubei Academy of Agricultural Sciences, Wuhan 430070, Hubei, China; 3International Research Center for Animal Disease, Ministry of Science and Technology of the People’s Republic of China, Wuhan 430070, Hubei, China

**Keywords:** *Mycobacterium tuberculosis*, natural compound, enniatin A1, bactericidal, synergy, membrane potential

## Abstract

*Background*: Tuberculosis remains a global disease that poses a serious threat to human health, but there is lack of new and available anti-tuberculosis agents to prevent the emergence of drug-resistant strains. To address this problem natural products are still potential sources for the development of novel drugs. *Methods*: A whole-cell screening approach was utilized to obtain a natural compound enniatin A1 from a natural products library. The target compound’s antibacterial activity against *Mycobacterium tuberculosis* (*M. tuberculosis*) was evaluated by using the resazurin reduction micro-plate assay (REMA) method. The cytotoxicity of the compound against Vero cells was measured to calculate the selectivity index. The intracellular inhibition activity of enniatin A1 was determined. We performed its time-kill kinetic assay against *M. tuberculosis*. We first tested its synergistic effect in combination with the first and second-line anti-tuberculosis drugs. Finally, we measured the membrane potential and intracellular ATP levels of *M. tuberculosis* after exposure to enniatin A1. *Results*: We identified enniatinA1 as a potential antibacterial agent against *M. tuberculosis*, against which it showed strong selectivity. Enniatin A1 exhibited a time-concentration-dependent bactericidal effect against *M. tuberculosis*, and it displayed synergy with rifamycin, amikacin, and ethambutol. After exposure to enniatinA1, the membrane potential and intracellular ATP levels of *M. tuberculosis* was significantly decreased. *Conclusions*: Enniatin A1 exhibits the positive potential anti-tuberculosis agent characteristics.

## 1. Introduction

Tuberculosis remains a global infectious disease that poses a serious threat to human health, causing about 10 million new cases and more than 1.5 million deaths in 2017 [1]. According to the World Health Organization (WHO) data, about one-third of the world’s population are under latent infection of tuberculosis, and they carry *M. tuberculosis* and act as a source of mobile infection source, posing great challenges for the prevention and treatment of tuberculosis [2,3]. In recent years, the wide use of anti-tuberculosis drugs has resulted in the emergence of drug-resistant strains, and the problem of drug resistance of *M. tuberculosis* has become increasingly severe [4]. Drug screening efforts have been made to detect new antibiotics or compound structures with new antibacterial mechanism by whole-cell or phenotypic high-throughput screening methods and to evaluate the anti-tuberculosis activity of natural products from large compound libraries [5,6,7]. Interestingly, as the latest anti-tuberculosis drug, a lead compound diarylquinoline (TMC207), a precursor of bedaquiline, has been discovered in recent decades by using a similar approach [8]. Bedaquiline is an effective drug against both wild-type and drug-resistant *M. tuberculosis* and it has been approved by the Food and Drug Administration (FDA) in 2012 for curing multidrug-resistant tuberculosis (MDR-TB) [9].

The integrity of the bacterial cell membrane is essential for bacteria to maintain their biological functions. Maintaining the electrochemical gradient of the cell membrane is necessary for providing impetus for the transmembrane transport of proteins and small molecules [10]. Most existing antibiotics target macromolecular biosynthesis. However, strains are prone to develop resistance against antibiotics. In order to reduce the emergence of resistant strains, new antibiotics with new antibacterial mechanisms of action are urgently needed, but it is difficult to detect such compounds [11]. In addition, the targeted destruction of cell membranes may be promising way for developing new drugs. Antimicrobial peptides were reported to kill bacteria by destroying the membrane integrity of bacteria [12]. However, antibiotics targeting *M. tuberculosis* cell membranes have rarely been reported. Therefore, detecting some lead compounds capable of destroying the function of *M. tuberculosis* cell membrane may be a promising research topic. 

Rifampicin is a classical anti-tuberculosis drug, which plays a crucial role in fighting against tuberculosis. It was well reported that up to now that the ‘click chemistry’ approach has provided a number of useful and fast synthetic methods for modifying biological compounds [13,14]. ‘Click chemistry’ was reported to have been successfully applied to the modification of rifamycins [14]. After many active rifamycin antibiotics had been modified, they exhibited better anti-tuberculosis activity [15]. Moreover, the form of rifamycin in solution is closely related to its biological activity. Several previous studies have confirmed the formation of zwitterionic structures and their derivatives in rifamycin [14]. The improvement of water solubility and lipophilicity enhanced rifamycins’ biological potency [15]. In addition, it was reported that proton transfer processes within rifampicin analogs influenced their biological activity [14,16]. Currently, some new advances has been made on the anti-bacterial mechanism of rifamycin, and recent studies on the mechanism of rifampicin at the molecular level have shown that the problem with use of this antibiotic is associated with the limited overcrossing the natural cell barriers and the enhanced affinity to efflux pump systems [17].

Natural compounds have been widely found in the environment, and they exist in large quantity with a wide array of biological activities. These natural compounds also are important source for antibiotics exploitation. Streptomycin is a classic antibiotic derived from natural products, and it has had a great impact on tuberculosis treatment [18]. Based on it, we utilized a whole-cell high-throughput screening approach to obtain the natural compound enniatin A1 (Figure 1) from a library of more than 6000 natural products. Enniatin A1 was found to be effective against both *M. tuberculosis* H37Rv and *M. bovis* BCG. Enniatins are mainly a mixture of secondary metabolites of cyclohexadepsipeptides isolated from *Fusarium* species fungi, with antifungal, antibacterial, and anticancer effects [19,20,21]. However, their antibacterial mechanisms of action remain unknown. Moreover, as early as the 1960s, a previous study reported that enniatins exhibited ionophoric activity [22]. More evidence indicated that enniatins (including enniatin A1) bound to the cell membranes to form a cation-selective pore, and subsequently formed a complex with monovalent metal cations to disrupt the transmembrane transport of ions, and to affect the concentration of intracellular metal cations, eventually destroying cellular physiological functions [19,23]. Thus, the antibacterial activity of enniatin A1 may be related to its ionophore properties.

## 2. Results

### 2.1. EnniatinA1 Was Active Against M. Tuberculosis in vitro and Intracellular

As shown in Table 1, the MICs of the compound against a range of gram-positive and gram-negative strains were determined, and the MIC of the compound against *M. tuberculosis* H37Rv and *M. bovis* BCG was 1.0 μg/mL and 2.0 μg/mL, respectively. The compound showed better selectivity to *M. tuberculosis*. The compound also showed significant inhibitory activity against *M. tuberculosis* in macrophage, and at a compound concentration of 8 μg/mL, intracellular *M. tuberculosis* was observed to be reduced by about 1 log10 CFU, compared with the control group (Figure 2A). 

In addition, in this experiment, Vero cells were used to assess the potential cytotoxicity of the compound. For Vero cells IC_50_ value of the compound were >64 μg/mL (Figure 2B). This result indicated that the compound had a high selectivity for the *M. tuberculosis* H37Rv (MIC = 1.0 μg/mL) in contrast with the measured cell with a selectivity index value (IC_50_/MIC) above 64.

### 2.2. Enniatin A1 Was Able to Kill M. Tuberculosis

The bactericidal effect of enniatin A1 against *M. tuberculosis* in vitro was time-concentration dependent. We determined the bactericidal activity of enniatin A1 against *M. tuberculosis* within the concentration range from 4 to 64 μg/mL. The bacteria were reduced by 1 log10 CFU (Figure 3) after 2 day of treatment with a compound concentration of 4 μg/mL. At the concentration of 64 μg/mL, on day 8, the compound had killed at least 4 log10 CFU, indicating a significant bactericidal effect of the compound against *M. tuberculosis*.

### 2.3. Antimicrobial Activity of Enniatin A1 in Combination with Classical Anti-Tuberculosis Drugs

In order to determine the antibacterial activity of enniatin A1 combined with classical anti-tuberculosis drugs, the chessboard method was used to perform drug interaction experiments. In this experiment, the MICs of rifampin (RIF), isoniazid (INH), ethambutol (EMB), amikacin (AMK), and enniatin A1 against *M. bovis* BCG were 0.00098 μg/mL, 0.0625 μg/mL, 0.125 μg/mL, 0.0625 μg/mL, and 2.0 μg/mL, respectively. As shown in Table 2, the combination of enniatin A1 with RIF, EMB, and AMK exhibited significant synergism (∑FIC < 0.5). In combination with EMB and AMK, enniatinA1 showed 16-fold and 8-fold reduction in MICs compared with their initial MICs, respectively. While combined with RIF, enniatinA1 showed best antibacterial activity, its MIC was reduced 128-fold compared with the MIC when enniatin A1 was used alone. However, compared with the other three drugs, INH showed no interaction (0.5 < ∑FIC < 4) in combination with enniatin A1. The combination of INH and RIF as a positive control showed strong synergistic action. Importantly, no antagonistic effect was observed when enniatin A1 was combined with classical anti-tuberculosis drugs, but their anti-tuberculosis activities were increased.

### 2.4. Enniatin A1 Disrupted Membrane Potential and Affected ATP Synthesis

Enniatin A1 has been reported to bind with alkaline metal cations and participate in the transmembrane transport of metal cations. Based on this, we measured the concentration of metal ions in the bacteria treated with enniatin A1 and found that only potassium ion concentration was significantly decreased (Figure 4). Therefore, we speculated that enniatin A1 might interfere with the membrane potential of *M. tuberculosis*. To verify this speculation, we used the fluorescent cyanine dye diethyloxacarbocyanine iodide (DiOC_2_) to label *M. bovis* BCG and *M. tuberculosis* H37Ra so as to determine the effect of enniatin A1 on membrane potential. DiOC_2_ is a fluorescent dye without membrane permeability, it can combine with the nucleic acid of bacteria, and the fluorescence will shift from green to red with the increase of membrane potential. Treatment with enniatin A1 resulted in a rapid and concentration dependent decrease of the membrane potential within an hour with the decreasing ratio of red to green fluorescence (Figure 5A). The oxidative phosphorylation uncoupler carbonyl cyanide 3-chlorophenylhydrazone (CCCP) was used as the positive control to dissipate the membrane potential. In addition, the decrease of the membrane potential resulted in the reduction in the proton motive force (PMF) of the cell membrane, which in turn affected the production of intracellular ATP. Therefore, we measured the level of intracellular ATP after exposure to enniatin A1. This result showed that the levels of intracellular ATP in *M. bovis* BCG were dramatically decreased in a concentration-dependent manner after exposed to different concentrations of enniatin A1 (Figure 5B). The effect of enniatin A1 on intracellular ATP production was consistent with effect of the bedaquiline in positive control group. In order to further observe the morphology and integrity of the cell membrane, we used scanning electron microscopy to analyze the submicroscopic structure of the bacteria exposed to enniatin A1.

### 2.5. Enniatin A1 didn’t Alter the Morphology and the Membrane Permeability of M. tuberculosis

In the abovementioned experiments, we have confirmed that enniatin A1 reduced the membrane potential of *M. tuberculosis*. To investigate whether enniatin A1 disrupted the integrity of membranes, we detected membrane permeability by labeling *M. bovis* BCG with a membrane-impermeable fluorescent dye SYTOX green. The results showed that neither a high concentration (32 μg/mL) nor a low concentration (4 μg/mL) of the compound disrupted the permeability of the cell membrane (Figure 5C). In order to further observe the morphology and integrity of the cell membrane, we observed the submicroscopic structure of the *M. bovis* BCG exposed to enniatin A1 by scanning electron microscopy, and found that the bacteria were intact and the cell membranes were not damaged (Figure 5D). These data indicated that enniatin A1 killed *M. tuberculosis* by affecting bacterial membrane potential and energy metabolism without disrupting membrane integrity.

## 3. Discussion

With the extensive use of anti-tuberculosis drugs, the problem of drug resistance of tuberculosis has become increasingly severe, posing serious challenges to existing methods of tuberculosis control. Moreover, with the emergence of multi-drug resistant and extensive drug-resistant strains, existing treatments cannot prevent the spread of tuberculosis [11]. For control the prevalence of this disease, new anti-tuberculosis drugs that can be used in combination with existing drugs to shorten treatment time are urgently needed [11]. Although, the development of a new antibiotic involve considerable time and effort with the low success rate [24], the attempts to develop new drugs have not stopped. Natural products remain a rich source of new drugs [25], and despite the current reduced interest in natural compounds, natural products, derivatives of natural products, and related lead compound structures remain potential sources of novel drugs [26]. It is worth noting that there are a wide variety of natural products in natural environment, which have a wide range of biological activities. And they also play an irreplaceable role in the process of fighting against diseases [27,28].

In this study, we found that ennaitin A1 exhibited significant bactericidal against *M. tuberculosis* H37Rv and *M. bovis* BCG, without affecting other tested strains. This result indicated it had a unique selectivity for *M. tuberculosis*. This antibiotic also exhibited a killing effect on *M. tuberculosis*. Moreover, enniatin A1 has been reported to be an antibiotic with ion carrier characteristics [23]. Currently, ionophore antibiotics are mainly used in livestock and veterinary industry for anti-parasite treatment [29]. The cytotoxicity of these antibiotics and poor selectivity in turn limits their further development into antibacterial agents. However, with the development of molecular biology, enniatins have been developed into an antibacterial agent for the treatment of upper respiratory tract infections [30]. In recent years, the special anticancer activity of enniatins has also attracted people’s attention, and its synergistic effect against cervical cancer has been reported in combination with the anti-tumor drug sorafenib [31]. Therefore, in any case, the importance of natural products should not be ignored, and many compounds like to enniatin A1 have exhibited have huge prospects.

Interestingly, in addition to in vitro bacteriostatic and bactericidal effects, enniatin A1 also showed synergistic effects against *M. tuberculosis* in combination with first and second-line anti-tuberculosis drugs, especially rifamycin. However, on which way such significant anti-tuberculosis potency improvement was achieved between enniatin A1 and rifamycin is still unknown. Recent research revealed that rifamycin exhibited three forms in solutions: A-, B-, and C-type [17]. The specific intermolecular interaction between water and rifamycin antibiotics resulted in the generation of more lipophilic structure (C-type) or more hydrophilic structure (A-type) during the transport of the antibiotic to the molecular target-RNA polymerase [17]. In addition, the equilibrium between A-and C-type conformers in rifamycin improves their adaptation to the changing nature of bacteria cell membranes, especially to efflux pump systems [17]. An interaction between enniatin A1 and rifampicin can contribute to the formation of a more lipophilic structure of rifampicin in solution, which could allow rifampicin to overcross the natural cell walls of *M. tuberculosis* strains. The interaction may affect the affinity of rifamycin antibiotics for efflux pump systems. Our further experiments demonstrated that after exposure to enniatin A1, the membrane potential and intracellular ATP levels of *M. tuberculosis* H37Ra and *M. bovis* BCG were significantly decreased. Maintaining the stability of bacterial cell membrane function is necessary to ensure cell homeostasis, and destroying bacterial cell membrane function and energy metabolism will seriously affect the survival of bacteria [32]. Our results suggest that enniatin A1 may be good adjuvant for improving efficacy of rifampicin against Mycobacteria strains. Moreover, antibiotics targeting membrane function may be able to reduce the emergence of resistant strains.

## 4. Materials and Methods 

### 4.1. Drugs and Chemicals Used in This Study

Enniatin A1 was purchased from TOKU-E (Bellingham, WA, USA). Bedaquiline was from TargetMol (Shanghai, China). INH, AMK, EMB, and RIF were from Selleck (Shanghai, China).

### 4.2. Bacterial Strains and Eukaryotic Cell Lines 

*M. tuberculosis* H37Rv ATCC27294, *M. tuberculosis* H37Ra ATCC25177, *M. bovis* ATCC19210, *M. bovis* BCG ATCC35737, and *M. smegmatis* mc^2^ 155 ATCC700044 were grown in Middlebrook 7H9 broth (BD, New York, NJ, USA) supplemented with 10% oleic-albumin-dextrose-catalase (OADC, BD, New York, NJ, USA ) and 0.2% glycerol (Sigma, Saint Louis, MO, USA), 0.05% Tween 80 (Amresco, Houston, TX, USA) or on 7H11 agar plates supplemented with 0.5% glycerol and 10% OADC. The other strains including *Escherichia coli* ATCC25922, *Staphylococcus aureus* ATCC75923, *Listeria monocytogenes* ATCC19115, *Klebsiella pneumonia* CTCC46117, and *Pseudomonas aeruginosa* ATCC9027 were cultured in LB broth with shaking. Vero cells (ATCC CCL-81) and THP-1 macrophages (ATCC TIB-202) were obtained from our laboratory and grown in Dulbecco’s modified Eagle’s medium (DMEM, Gibco, Waltham, MA, USA) and RPMI-1640 medium (Gibco, Waltham, MA, USA) supplemented with 10% fetal bovine serum (FBS, Gibco, Waltham, MA, USA) at 37 °C with 5% CO_2_, respectively.

### 4.3. Determination of MIC and Measurements of Synergy 

Minimum inhibitory concentrations (MICs) against *M. tuberculosis* and other strains were determined using the resazurin reduction micro-plate assay (REMA) method as previously described [33]. Briefly, bacteria were grown to log phase (optical density 600 nm [OD600] of 0.6 to 0.8) and diluted to an OD600 of 0.05. Two-fold serial dilutions of test compound were prepared in 96-well micro-plate containing 100 μL diluted solutions of bacteria. The last row of the plate was used as a drug-free control. RIF was used as antibiotic control. Plates were incubated at 37 °C for the required incubation time (for example 7 days for *M. tuberculosis* at 37 °C, only 24 h for other strains). The 30 μL of 0.01% resazurin was added to each well and incubated for 24 h at 37 °C, and color variation was evaluated. A color change from blue to pink indicates the bacterial growth. The lowest drug concentration to stop this color change was defined as MIC. The combination effect of enniatinA1 with RIF, INH, EMB, or AMK against *M. bovis* BCG was determined using the checkerboard method as described previously [34]. In brief, the MICs of the tested compound and anti-tuberculosis drugs against *M. bovis* BCG were evaluated separately. Diploid serial dilutions of compounds were prepared in 96 well micro-plate. *M. bovis* BCG at log phase was diluted to an OD600 of 0.05. A 50 μL volume of the diluted cells were added to each well of 96 well micro-plate. The micro-plate was incubated at 37 °C for seven days, and resazurin solution was added to each well. MIC was defined as described above. The fractional inhibitory concentration (FIC) of each compound was calculated as the MIC of the compound in combination divided by the sum of MICs of compounds used alone. The fractional inhibitory concentration index (FICI) represents the sum of the FIC of the two compound components [35,36].

### 4.4. Cytotoxicity for Vero Cells 

A volume of 100 μL Vero cells (1 × 10^4^ cells/well) were seeded into a 96-well plate and incubated with different concentration of compound at 37 °C with 5% CO_2_ for 24 h. Then each well was added 10 μL with WST-1 (Beyotime, Shanghai, China) solution, and the plate was incubated in a cell incubator for 2 h and shaken for one minute before reading fluorescence with a fluorescence micro-plate reader (SPARK 10M, TECAN, Männedorf, Switzerland) at OD_450_. The minimum concentration at which at least 50% cell growth was inhibited was defined as IC_50_.

### 4.5. Kill Kinetic Assay 

*M. bovis* BCG cultures were prepared in 7H9 broth at an OD_600_ of 0.05 and then divided to six 10 mL aliquots and supplemented with a final concentration of 4 μg/mL, 8 μg/mL, 16 μg/mL, 32 μg/mL, and 64 μg/mL of enniatin A1, respectively, including a no drug control. At day 0, 2, 4, 8 and 12, cultures were ten-fold serially diluted and plated in 7H11 agar plates. All the plates were incubated at 37 °C for 3 weeks, and then the CFUs were calculated.

### 4.6. Activity of EnniatinA1 against Intracellular M. Tuberculosis 

THP-1 cells were seeded into 24-well plates and incubated with a final concentration of 100 nM phorbol 12-myristate 13-acetate (PMA) for 48 h to induce cell differentiation into macrophages. Cells were infected with *M. tuberculosis* H37Rv at a multiplicity of infection (MOI) 1:10. After 4 h of infection, the extracellular bacteria were removed by washing with phosphate-buffered saline (PBS, pH7.2). The compound were prepared in the RPMI-1640 medium with a final concentration of 8 μg/mL and 32 μg/mL. Culture was replaced with 1 mL of the compound-containing media. Plates were incubated at 37 °C with 5% CO_2_ for 24 h. And then, the infected macrophages were lysed by adding 1 mL 0.025% Triton-100. Diluting the cell lysate and plating on 7H11 agar plates to calculate CFUs. Triplicate cultures of each strain were analyzed during a single experiment and the experiments were repeated at least three times.

### 4.7. Measurement of Intracellular Ion Content 

Intracellular ion content was measured as described previously [37,38]. Briefly, a volume of 10 mL logarithmic phase of *M. bovis* BCG was collected and washed with PBS for two times, the pellets were re-suspended in equal volume of PBS containing 4 μg/mL enniatinA1, and then incubated in incubator for 24 h. The culture was centrifuged at 12,000 rpm for 5 min. The wet pellet weight was measured and bacteria were chemically lysed using Bugbuster (Novagen, Madison, WI, USA) according to the manufacturer’s instructions. Bacteria were re-suspended in Bugbuster solution and incubation on a shaker at room temperature for 24 h. Total protein for each sample was measured by using NanoDrop ND-1000 spectrophotometer (NanoDrop, Waltham, MA, USA) according to the manufacturer’s instructions. Sample was diluted 100-fold in 2% molecular grade nitric acid to a total volume of 10 mL. Samples were analyzed by inductively coupled plasma mass spectrometry (ICP-MS, 802-MS, Palo Alto, CA, USA), and the results were corrected using the appropriate buffers. Triplicate cultures of each strain were analyzed during a single experiment and the experiments were repeated at least three times.

### 4.8. Determination of Membrane Potential

The membrane potential was determined by using a Baclight bacterial membrane potential kit (Invitrogen, Waltham, MA, USA). *M. tuberculosis* H37Ra and *M. bovis* BCG were centrifuged and suspended in 1 mL PBS (pH 7.2), and then they were stained with 30 μM DiOC_2_(3) (3,3′-diethyloxa-carbocyanine iodine fluorescent dye) for 30 min at 37 °C. After staining, the cells were centrifuged and re-suspended in the same buffer to an OD600 of 0.1. Afterwards, 100 μL of labeled bacteria were added into 96-well plates, and treated with 2 μg/mL, 4 μg/mL, and 8 μg/mL of enniatinA1. With 5 μg/mL CCCP as positive control and 2% dimethyl sulfoxide (DMSO) as a solvent control in the experiment. The fluorescence of green and red were measured using a fluorescence micro-plate reader (TECAN SPARK 10 M) at excitation 488 nm, and emission 515 nm/610 nm, respectively. The ratio of red/green (610 nm/515 nm) fluorescence intensity is relative to the strength of membrane potential. Triplicate cultures of each strain were analyzed during a single experiment and the experiments were repeated at least three times.

### 4.9. Determination of Membrane Permeability

We used the cyanine dye SYTOX green (Ex/Em: 488 nm/525 nm) to detect bacterial cell membrane permeability. This dye doesn’t have membrane permeability, but it can easily permeate the damaged cell’s plasma membrane. When the cytoplasmic membrane integrity is destroyed, the dye can cause a great enhancement in fluorescence intensity by binding to nucleic acids. In this assay, *M. bovis* BCG was grown to log phase and then diluted to an OD600 of 0.1. Culture was incubated with 3 μM SYTOX green in the dark for 30 min in incubator and added to a 96-well microplate. Background fluorescence was measured before addition of compound. Then, the fluorescence was measured every 5 min for 1 h after treatment with 4 μg/mL, 8 μg/mL, 16 μg/mL, and 32 μg/mL of compound. The 2% DMSO was used as a solvent control and culture without drug as a blank control. Triplicate cultures of each strain were analyzed during a single experiment and the experiments were repeated at least three times.

### 4.10. Scanning Electron Microscopy of M. Bovis BCG 

1.0 × 10^8^ CFUs bacteria were treated with 4 μg/mL enniatin A1 for 24 h. Bacteria were re-suspended in 2.5% glutaraldehyde for 2 h at room temperature. Samples were washed with 0.1 M phosphate buffer (pH 7.4) for three times and fixed with 0.1 M phosphate buffer (pH 7.4) containing 1% citric acid for 2 h at room temperature, and then the samples were washed with 0.1 M phosphate buffer (pH 7.4) for another three times. Samples were dehydrated with gradient absolute ethanol and isoamyl acetate. The dehydrated samples were dried in a critical point dryer (K850, Quorum, Laughton, East Sussex, UK). The dried samples were placed on a double-sided adhesive tape of a conductive carbon film and placed in an ion sputtering apparatus (MSP-2S, IXRF, Austin, TX, USA) for gold spraying. The samples were observed with a scanning electron microscope (SU8100, HITACHI, Tokyo, Japan).

### 4.11. Quantification of Intracellular ATP 

Intracellular ATP was measured by using the BacTiter-Glo Microbial Cell Viability Assay Kit (Promega, Madison, WI, USA) according to the manufacturer’s instructions. Briefly, *M. bovis* BCG at exponential phase was diluted to an OD600 of 0.05 and exposed to enniatinA1 at 4 μg/mL, 8 μg/mL, and 16 μg/mL for 24 h, mixed with an equal volume of BacTiter-Glo reagent, and incubated for 5 min in the dark. With a 10 μg/mL of bedaquiline as positive control. Luminescence was detected by using a fluorescence micro-plate reader (TECAN SPARK 10 M). Triplicate cultures of each strain were analyzed during a single experiment and the experiments were repeated at least three times.

## Figures and Tables

**Figure 1 molecules-25-00038-f001:**
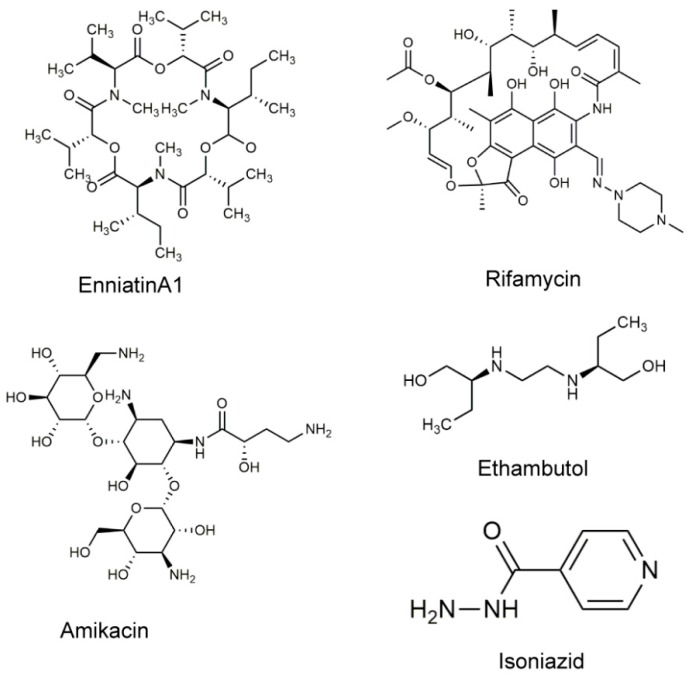
Structures of the studied bioactive molecules.

**Figure 2 molecules-25-00038-f002:**
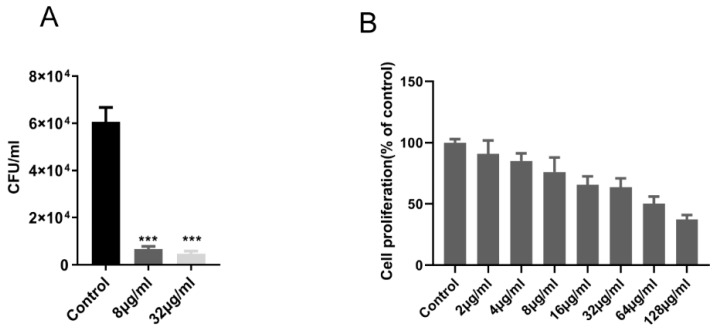
Enniatin A1 was active against intracellular *M. tuberculosis* with mild cytotoxicity to Vero cell line. (**A**) *M. tuberculosis* H37Rv infected THP-1 macrophages were exposed to enniatin A1. CFUs were calculated on 7H11 agar plates at the appointed days post-infection. (**B**) Vero cells were exposed to different concentration of enniatin A1 for 24 h, and cell viability was detected by adding of WST-1 reagent. Data were presented as mean ± SD, statistical analysis was performed unpaired via Student’s t-test (*** *p* < 0.0001).

**Figure 3 molecules-25-00038-f003:**
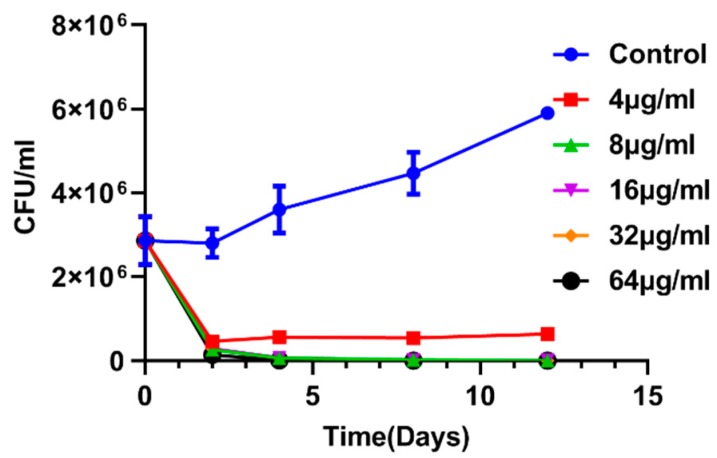
Time-concentration dependent kill curve of enniatin A1. *M. bovis* BCG at exponential phase was exposed to different concentration of enniatin A1 for 12 days at 37 °C. At the indicated time, CFUs were counted. Data from three technical replicate experiments are presented as mean ± SD.

**Figure 4 molecules-25-00038-f004:**
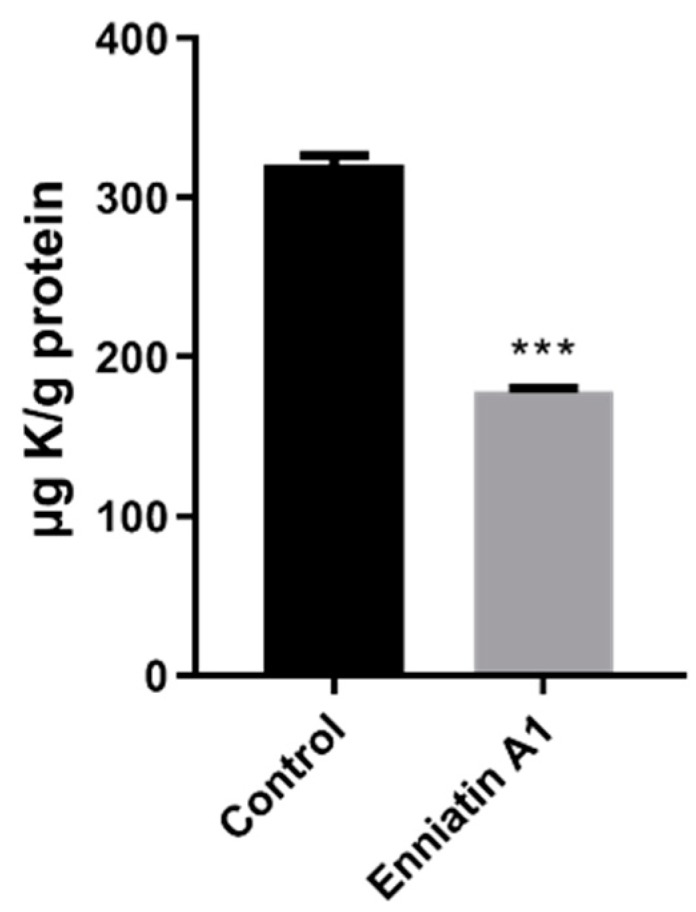
Exposure to enniatin A1 reduced the level of intracellular potassium in *M. tuberculosis*. Mid-exponential phase of *M. bovis* BCG was exposed to 2.0 μg/mL enniatin A1.The level of K^+^ in *M. bovis* BCG cells was measured by inductively coupled plasmon resonance atomic absorption spectrometry (ICP-MS). *** *p* < 0.0001.

**Figure 5 molecules-25-00038-f005:**
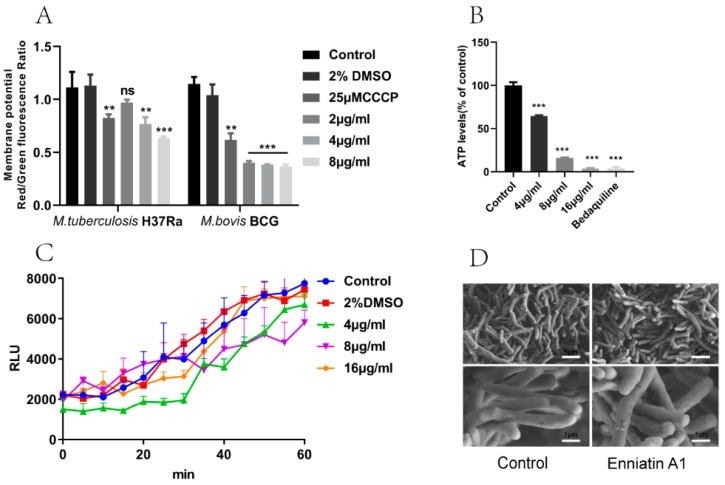
Mechanism of enniatin A1 against *M. tuberculosis*. (**A**) Effect of enniatin A1 on membrane potential of *M. tuberculosis* H37Ra and *M. bovis* BCG. The cultures at exponentially phase were exposed to 2–8 μg/mL enniatin A1. We determined the membrane potential by labeling bacteria with DiOC_2_. The (*v*/*v*) % of DMSO was consistent with the highest concentration of enniatin A1 (below 2%). (**B**) Effect of enniatin A1 on mycobacterial ATP levels. *M. bovis* BCG were treated with the indicated concentration of ennaitin A1 and the ATP levels was measured by previously described methods. (**C**) Effect of enniatin A1 on mycobacterial membrane permeability to small molecular compounds. *M. bovis* BCG was exposed to different concentrations of enniatin A1 and incubated with cyanine dye SYTOX green at 37 °C. Membrane permeability was determined by monitoring the intracellular fluorescence signal accumulation. (**D**) Effect of enniatin A1 on mycobacterial morphology. Exponentially growing *M. bovis* BCG was treated with or without 4 μg/mL enniatin A1 for 24 h and observed by scanning electron microscopy. Data from three independent experiments were expressed as mean ± SD, statistical analysis was performed via unpaired Student t-tests (** *p* < 0.001, *** *p* < 0.0001).

**Table 1 molecules-25-00038-t001:** MICs of enniatin A1 against microorganisms.

Microorganisms	MIC (μg/mL)
*M. tuberculosis* H37Rv	1.0
*M. tuberculosis* H37Ra	2.0
*M. bovis*	2.0
*M. bovis* BCG	2.0
*M*. *smegmatis* mc^2^ 155	8.0
*E. coli*	>100
*S. aureus*	>100
*p. aeruginosa*	>100
*K. pneumonia*	>100
*L. monocytogenes*	>100

**Table 2 molecules-25-00038-t002:** Combination of enniatin A1 and anti-tuberculosis drugs against *M. tuberculosis*.

Drug	Combination	MIC (μg/mL)		** FIC	*** ∑FIC	Remarks
		Alone	Combination			
RIF		0.00098	0.000045	0.0459	0.2959	
INH		0.0625	0.0156	0.25		Synergism
RIF		0.00098	0.00012	0.125	0.1328	
* E-A1		2.0	0.0156	0.0078		Synergism
INH		0.0625	0.00049	0.0078	1.0078	
E-A1		2.0	2.0	1.0		No interaction
EMB		2.0	0.125	0.0625	0.1250	
E-A1		2.0	0.125	0.0625		Synergism
AMK		1.0	0.0625	0.0625	0.1875	
E-A1		2.0	0.25	0.125		Synergism

* E-A1: enniatin A1; ** FIC_drugA_ = MIC_drugA_ in combination with drug B/MIC_drugA_, FIC_drugB_ = MIC_drugB_ in combination with drug A/MIC_drugB_; *** ∑FIC = FIC_drugA_ + FIC_drugB_, ∑FIC ≤ 0.5, 0.5 < ∑FIC < 4.0, and ∑FIC ≥ 4.0 represent synergy, no interaction, and antagonism, respectively.

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
