# Peer review of "Enniatin A1, A Natural Compound with Bactericidal Activity against Mycobacterium tuberculosis In Vitro"

_molecules, 2019, doi:10.3390/molecules25010038_

Round 1

Reviewer 1 Report

Dear collegues,

in general I like your research.

Some general comments:

1) the paper needs an extensive language correction

2) some formal reviions of text, and quality of graphics can be improved

3) I miss some standard control compounds for the experiments you have done.

Author Response

Response to Reviewer 1 Comments

Point 1: the paper needs an extensive language correction

Response 1: I am very grateful to your comments for our manuscript. According with your advice, we examined and amended the language in our manuscript. And my manuscript has been polished by an experienced linguistics professor. Here below is a language polishing statement:

This is a statement for Language polishing. I want to tell you that the language of the manuscript has been polished by an experienced linguistics professor Ping Liu, who is the dean of the English Academic Writing Center of a prestigious University. This center is responsible for providing professional English editing or language polishing service for science papers written in English. Professor Liu has been engaged in language polishing service for more than ten years and teaching doctoral students the course of Academic English Writing for several years on end in a prestigious University. Although she is non-English native speaker, but she has ever been to USA and UK as a visiting scholar staying in these two English speaking countries respectively for one year.

A face to face language polishing has been conducted. Professor Liu and I, the first writer, worked on this paper's language for one entire day. Major grammatical mistakes in the manuscript have been corrected. Some originally not understandable or poorly expressed sentences or even paragraphs have been rewritten. We have done whatever we could to improve the language expression. We are sure that there are no problems for target readers to understand the language expressions of this paper. I wonder whether you could give us some specific examples if you find any language expression unacceptable in the manuscript. Sure, the language is still far from perfect and there is much room to be improved. Since we are not sensitive to the minor linguistic mistakes and maybe not be familiar with academic writing norms in specific discipline as non-English native writers and a novice. I wonder if you could even do me a big favor to correct those improper expressions and mistakes identified by you. I will greatly appreciate your linguistic help as an English native speaker. I will remember your help and support in the growth of a novice writer. 

Point 2: some formal reviions of text, and quality of graphics can be improved

Response 2: According with your advice, we have carefully examined our manuscript and amend our manuscript text form. Revised portion are marked in the paper. To improve the quality of the figures, we have re-created our graphics using the software of GraphPad prism8 and Adobe Illustrator and have made some changes in the revised manuscript. Thanks for all your feedback and suggestions.

Point 3: I miss some standard control compounds for the experiments you have done.

Response 3: Thanks for you have made a very good advice here, we are very sorry for our negligence of some of the results lack the control of standard drugs in the experiments. We have been aware of the inadequacy of our work, and I will improve our scientific research level in accordance with your suggestions and make more achievements in the future work. Once again, thank you very much for your comments and suggestions on our paper.

Reviewer 2 Report

Tan et al. submitted manuscript (molecules-656381) dealing with improving of antitubercular potency of standard antimicrobials after addition of the proposed adjuvant - Enniatin A1. Among studied antimicrobials is rifampicin (RIF), which has been recently studied toward this direction (please see articles: Pyta et al. Eur J Med Chem, 2019, 167, 96-104; Pyta et al. Eur J Med Chem 2014, 84, 651-676), however, Authors "do not see" these findings (lack of citations of current literature in the Introduction). Recent article, dealing with the mechanism of rifampicin (known antitubercular agent) on molecular level revealed, that the problem with the use of this antibiotic is concerned with the limited overcrossing the natural cell barriers and the enhanced affinity to efflux pump systems (see Pyta et al. ACS Infect. Dis. 2019, 5, 10, 1754-1763 entitled: "Specific Interactions between Rifamycin Antibiotics and Water Influencing Ability To Overcome Natural Cell Barriers and the Range of Antibacterial Potency" - also not cited article!). Main stream of Authors' discussion in the manuscript is focused on the fact that, after the addition of enniatin A (cyclic peptide - classical ionophore) which is adjuvant to standard antimicrobials (e.g. rifampicin, RIF), the antimycobacterial potency of the antibiotic seems to be enhanced (the most interesting result among all reported ones). It is not clear, in opinion of Reviewer, however, on which way such spectacular antibacterial potency improvement is achieved? Authors must discuss shortly this issue in the light of the mechanism of Rif published in ACS Inf Dis 2019, 5, 10, 1754-1763. Do Authors postulate some interactions between RIF and Enniatin A or formation by Enniatin A some channel within cell barriers or increased fluidity of the cell membranes of M. tuberculosis? Authors generally postulated that cell membrane integrity is conserved after use of Enn A and that the altered parameter is the membrane potential, so they postulated that Enn A acts as typical ionophore in the presence of e.g. RIF. For RIF the two structural forms are observed in solutions and in solid: non-charged or zwitterionic. Thus, here question arises: what is the influence of the changed membrane potential via Enn A on the form of rifampicin, which is crucial for transporting of the drug to the target i.e. bacterial RNA polymerases (Authors "forgot" cite many references in this topic...)? Some doubts also arise when one analyses values of MICs, determined for RIF before and after addition of Enn A (e.g. MIC 0.00012 ug/mL) - I think that they are determined on the accurancy limit of the analytical method. It is also wort to check, whether other rifamycin like rifapentine behaves similarly to rifampicin in the presence of Enn A. In my opinion these results are worth publishing but only after major revision.

Author Response

Response to Reviewer 2 Comments

Point 1: Among studied antimicrobials is rifampicin (RIF), which has been recently studied toward this direction (please see articles: Pyta et al. Eur J Med Chem, 2019, 167, 96-104; Pyta et al. Eur J Med Chem 2014, 84, 651-676), however, Authors "do not see" these findings (lack of citations of current literature in the Introduction). Recent article, dealing with the mechanism of rifampicin (known antitubercular agent) on molecular level revealed, that the problem with the use of this antibiotic is concerned with the limited overcrossing the natural cell barriers and the enhanced affinity to efflux pump systems (see Pyta et al. ACS Infect. Dis. 2019, 5, 10, 1754-1763 entitled: "Specific Interactions between Rifamycin Antibiotics and Water Influencing Ability To Overcome Natural Cell Barriers and the Range of Antibacterial Potency" - also not cited article!)

Response 1: Thanks for your valuable and helpful comments. We are very sorry to have missed these interesting findings about rifampicin. According with your advice, we have carefully read these article and amended in our paper. We have cited current literature and have described the latest antibacterial mechanism of rifampicin on molecular level in the Introduction. (Page 2 lines 62-75)

Point 2: Authors' discussion in the manuscript is focused on the fact that, after the addition of enniatin A (cyclic peptide - classical ionophore) which is adjuvant to standard antimicrobials (e.g. rifampicin, RIF), the antimycobacterial potency of the antibiotic seems to be enhanced (the most interesting result among all reported ones). It is not clear, in opinion of Reviewer, however, on which way such spectacular antibacterial potency improvement is achieved? Authors must discuss shortly this issue in the light of the mechanism of Rif published in ACS Inf Dis 2019, 5, 10, 1754-1763.

Response 2: To be honest, we didn't think much about it before you raised the question in this work. We usually pay more attention to whether there is synergy against M. tuberculosis between the compound and anti-tuberculosis drugs, so we usually neglect the way in which they achieve such spectacular antibacterial effect. Thank you for pointing this out. We have carefully read these articles and have discussed this issue in the Discussion. (Page 7 line 224-234)

Point 3: Do Authors postulate some interactions between RIF and Enniatin A or formation by Enniatin A some channel within cell barriers or increased fluidity of the cell membranes of M. tuberculosis? Authors generally postulated that cell membrane integrity is conserved after use of Enn A and that the altered parameter is the membrane potential, so they postulated that Enn A acts as typical ionophore in the presence of e.g. RIF. For RIF the two structural forms are observed in solutions and in solid: non-charged or zwitterionic. Thus, here question arises: what is the influence of the changed membrane potential via Enn A on the form of rifampicin, which is crucial for transporting of the drug to the target i.e. bacterial RNA polymerases (Authors "forgot" cite many references in this topic...)?

Response 3: As you said, we initially thought it was enniatinA1 that formed a channel in the bacterial membrane, resulting in the accumulation of rifampicin in the bacteria and achieving a significant bactericidal effect. However, our subsequent experiments shown that enniatinA1 did not disrupt the membrane permeability of M. tuberculosis. Recent studies have shown that rifamycin revealed two types in solutions: zwitterionic and non-ionic. We hypothesized that the changed membrane potential via enniatinA1 may increase the lipophilicity of rifampicin, making it easier to penetrate the membrane of M. tuberculosis. The presence of rifamycin as zwitterions and especially double zwitterions with the intramolecularly transferred protons, improves water solubility and binding mode of the antibiotic to molecular target –RNA polymerase. Furthermore, molecular docking analyses of the obtained rifamycin derivatives as zwitterions at the binding site of bacterial RNA polymerase have shown key intermolecular interactions contributing to the stability of the enzyme.

Point 4: Some doubts also arise when one analyses values of MICs, determined for RIF before and after addition of Enn A (e.g. MIC 0.00012 ug/mL) - I think that they are determined on the accurancy limit of the analytical method.

Response 4: We are very sorry for causing some doubts to you here. In this work, we determined the MIC of RIF against M. bovis BCG based on the method of two-fold serial dilutions according with related articles. (Please see articles: Caroline, et al. mBio, 2018, 9, 01276-18; Ahmad, et al. Antimicrobial agents and chemotherapy, 2011, 55, 239-245; Odds, et al. The Journal of antimicrobial chemotherapy, 2003, 52, 1). Moreover, Caroline, et al. mBio, 2018, 9, 01276-18 entitled “Arylvinylpiperazine Amides, a New Class of Potent Inhibitors Targeting QcrB of Mycobacterium tuberculosis”, in this article, RIF shows a MIC 0.0008 ug/mL to M. bovis BCG. Once again, thank you very much for your comments and suggestions.

Point 5: It is also wort to check, whether other rifamycin like rifapentine behaves similarly to rifampicin in the presence of Enn A.

Response 5: Thanks for you have made a very good advice here, the point suggested by you are interesting and would provide valuable information to our study. Recently, rifapentine has been reported to enhance compound anti-tuberculosis effects in combination with new compound. (Please see articles: Thomson, et al. Antimicrob Agents Chemother, 2015, 1455-65). And we wish to finish it in our future work. Finally, special thanks to you for your good comments.

Round 2

Reviewer 1 Report

Hello,

dear authors, I see that you tried to improve the manuscript.

I now recommend to be published, however, I still see problems in the language. I believe, that your correctors are good scientists, but some language facts are still problematic. Please, try to go on through the manuscript once again, and use the service of a real native speaker. This will really improve the quality of the text.

With best wishes.

Author Response

Point 1: I now recommend to be published, however, I still see problems in the language. I believe, that your correctors are good scientists, but some language facts are still problematic. Please, try to go on through the manuscript once again, and use the service of a real native speaker. This will really improve the quality of the text. Response 1: We are very sorry for causing some doubts to you due to our carelessness, our manuscript has been polished by another experienced linguistic professor again. All revisions were clearly highlighted in red marked. Once again, thank you very much for your professional and helpful comments on our manuscript.

Reviewer 2 Report

Tian and Lu et al submitted revised manuscript (molecules-656381) which adresses all issues raised by reviewer. Introduction and Discussion were enriched by an important informations in the field. The topic of this article is stimulating for further studies on the detailed mechanism of action leading to synergic effect of rifampicin and enniatin A and should be well received by a broad, multidisciplinary audience (it should be certainly well-cited). Apart from this point, the authors achieved a spectacular result regarding the improvement of rifampicin activity toward M. tuberculosis and this is the main scientific advantage, therefore I recommend this valuable manuscript for publication in Molecules. Simultaneously, I recommend also some minor revisions, which should improve readibility of the manuscript.

Minor revisions:

1) One figure showing all structures with codes (chemdraw, chemsketch or other one types) of the studied bioactive molecules, including enniatin A1, rifampicin etc. should be placed in the manuscript. This manuscript is expected to be published in Molecules journal and should also adress the chemistry part.
2) line 75, page 2. is "many active rifamycins antibiotic..." maybe should be written "many active rifamycin antibiotics"?
3) line 184, page 5, is "Carbonyl" in the middle of the sentence - maybe should be written "carbonyl"?
4) line 283, page 7 is ""...and rifamycin still unknown." - maybe should be "...and rifamycin is still unknown"?
5) lines 290-292, page 7 - please consider to correct this text into: ""An interaction between enniatinA1 and rifampicin can contribute to formation of a more lipophilic structure of rifampicin in solution, which could allow rifampicin to overcome natural cell walls of M. tuberculosis strains. The interaction may also affect the affinity..."
6) lines 296-297, page 7 - please consider to correct this text into: "Our results suggest that enniatinA1 may be good adjuvant improving efficacy of rifampicin towards Mycobacteria strains."

Author Response

Point 1: One figure showing all structures with codes (chemdraw, chemsketch or other one types) of the studied bioactive molecules, including enniatin A1, rifampicin etc. should be placed in the manuscript. This manuscript is expected to be published in Molecules journal and should also adress the chemistry part. Response 1: Thanks for you have made a great advice here, we have added a figure included all structures of the studied antibiotics in our revised manuscript. (Line 90, page 3, figure 1) Point 2: line 75, page 2. is "many active rifamycins antibiotic..." maybe should be written "many active rifamycin antibiotics"? Response 2: Thank you very much for your correction. We have made corrections in our revised manuscript and marked it in red. (Line 66, page 2) Point 3: line 184, page 5, is "Carbonyl" in the middle of the sentence - maybe should be written "carbonyl"? Response 3: Thank you very much for your correction. We have made corrections in our revised manuscript and marked it in red. (Line 154, page 5) Point 4: line 283, page 7 is ""...C." – may be should be "...and rifamycin is still unknown"? Response 4: Thank you very much for your correction. We have made corrections in our revised manuscript and marked it in red. (Line 227, page 8) Point 5: lines 290-292, page 7 - please consider to correct this text into: ""An interaction between enniatinA1 and rifampicin can contribute to formation of a more lipophilic structure of rifampicin in solution, which could allow rifampicin to overcome natural cell walls of M. tuberculosis strains. The interaction may also affect the affinity..." Response 5: Thank you very much for your correction. We have made corrections in our revised manuscript and marked it in red. (Line 232-235, page 8) Point 6: lines 296-297, page 7 - please consider to correct this text into: "Our results suggest that enniatinA1 may be good adjuvant improving efficacy of rifampicin towards Mycobacteria strains." Response 6: Thank you very much for your correction. We have made corrections in our revised manuscript and marked it in red. (Line 240-241, page 8)
